# HOW CONFIDENT ARE VIDEO MODELS?
## EMPOWERING VIDEO MODELS TO EXPRESS THEIR UNCERTAINTY

## ABSTRACT

Generative video models demonstrate impressive text-to-video capabilities, spurring widespread adoption in many real-world applications. However, like large language models (LLMs), video generation models tend to *hallucinate*, producing plausible videos even when they are factually wrong. Although uncertainty quantification (UQ) of LLMs has been extensively studied in prior work, no UQ method for video models exists, raising critical safety concerns. To our knowledge, this paper represents the first work towards quantifying the uncertainty of video models. We present a framework for uncertainty quantification of generative video models, consisting of: (i) a metric for evaluating the calibration of video models based on robust rank correlation estimation with no stringent modeling assumptions; (ii) a black-box UQ method for video models (termed **S-QUBED**), which leverages latent modeling to rigorously decompose predictive uncertainty into its aleatoric and epistemic components; and (iii) a UQ dataset to facilitate benchmarking calibration in video models. By conditioning the generation task in the latent space, we disentangle uncertainty arising due to vague task specifications from that arising from lack of knowledge. Through extensive experiments on benchmark video datasets, we demonstrate that S-QUBED computes calibrated total uncertainty estimates that are negatively correlated with the task accuracy and effectively computes the aleatoric and epistemic constituents.

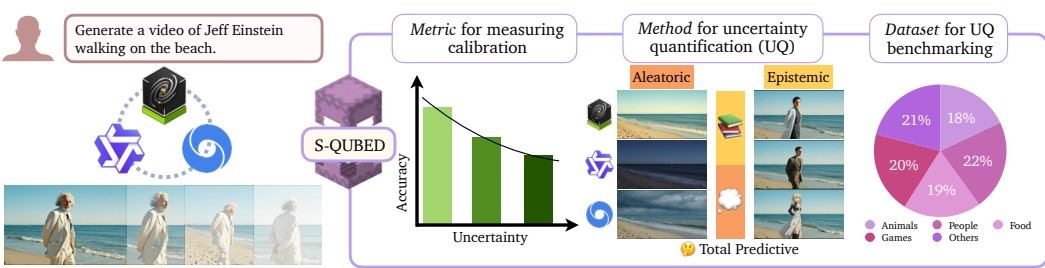

Figure 1: Video models are unable to express their uncertainty, posing a critical limitation especially in tasks where they lack requisite knowledge. Here, the video model generates an inaccurate video (showing Albert Einstein), when prompted to generate a video of Jeff Einstein. To this end, we introduce a *metric* for evaluating the calibration of video models, a *calibrated uncertainty quantification method* (S-QUBED) which uses latent modeling to disentangle aleatoric and epistemic uncertainty, and a *UQ dataset* for benchmarking calibration.

## 1 INTRODUCTION

Recent advances in video generation models have led to huge strides in their capabilities (DeepMind, 2025; NVIDIA et al., 2025). However, current text-to-video models tend to hallucinate, generating videos misaligned with the user intention, or disobeying physical laws. Despite this important limitation, existing video models are unable to express their own uncertainties, unlike LLMs, posing

a crucial safety concern. We illustrate hallucinations in video models in Figure 1. When prompted to generate a video of Jeff Einstein walking on a beach, the video model generates a video of Albert Einstein, an entirely different person, without expressing any doubt in its output. We aim to address this critical challenge by empowering video models to express their uncertainty.

Specifically, we propose a framework for uncertainty quantification of video models, consisting of three fundamental components: First, we introduce a *metric* for evaluating the calibration of video models that directly assesses the alignment of the uncertainty estimates with the accuracy of the video generation task. Our metric estimates the rank correlation between uncertainty and task accuracy to measure the calibration error.

Second, we derive *S-QUBED* (Semantically-Quantifying Uncertainty with Bayesian Entropy Decomposition), a black-box uncertainty quantification method for video generation models, preserving amenability to the ever-increasing set of closed-source video models. Our key insight is to quantify uncertainty with latent modeling, enabling the rigorous decomposition of predictive uncertainty into its aleatoric and epistemic components. By mapping the input text prompt to a latent space, S-QUBED effectively distinguishes between uncertainty arising from ambiguous prompts and uncertainty arising from the model's lack of knowledge. We demonstrate the calibration of S-QUBED's estimates across a broad variety of video generation tasks.

Third, we curate a *UQ dataset* of about 40K videos across diverse tasks to facilitate benchmarking UQ methods for video models. We generate the data using the open-source model Cosmos-Predict2 (NVIDIA et al., 2025). We hope that the dataset drives research on uncertainty quantification of video models.

## 2 RELATED WORK

**Uncertainty Quantification in Deep Learning.** Deep neural networks (DNNs) are generally difficult to interpret (Li et al., 2022), motivating the development of UQ methods to examine the trustworthiness of their predictions (Abdar et al., 2021). UQ methods in deep learning can be broadly categorized into: *training-free* and *training-based* methods, which constitute a majority of existing work. Training-free methods estimate uncertainty without modifying the model's architecture, training algorithm, or dataset, e.g., via perturbation techniques (Liu et al., 2024), dropout injection (Loquercio et al., 2020; Ledda et al., 2023), and test-time data augmentation (Ayhan & Berens, 2018; Wu & Williamson, 2024). In contrast, training-based methods impose specific architectural design choices to enable uncertainty quantification using Bayesian Neural Networks (BNN) and can be further classified into three categories: (i) variational inference, (ii) Monte-Carlo Dropout, and (iii) Deep Ensemble methods. Assuming that the parameters (weights) of learned models are random variables, BNN methods Kononenko (1989) apply Bayes' rule to estimate a posterior distribution over these parameters given a prior distribution. However, the exact application of Bayes' rule is typically intractable, giving rise to approximation techniques, e.g., variational inference (Zhang et al., 2018a), which approximates the posterior distribution using a parametric distribution; Monte-Carlo Dropout Gal & Ghahramani (2016), which samples from the posterior distribution by zeroing-out some weights; and Deep Ensembles (Lakshminarayanan et al., 2017), which train multiple independent models to represent the posterior distribution. Despite their success, traditional UQ methods in deep learning are computationally expensive, limiting their applications in large generative models, e.g., large language models (LLMs) and vision-language models (VLMs). UQ methods for LLMs/VLMs generally leverage internal activations of these models, or utilize similarity-based metrics or natural-language inference techniques for more efficient UQ (see (Shorinwa et al., 2025) for a detailed discussion).

**Uncertainty Quantification in Generative Image/Video Models.** Unlike DNNs and LLMs, UQ of generative image/video models has been relatively underexplored (Franchi et al., 2025). Prior work (Chan et al., 2024) extends Bayesian UQ techniques to denoising diffusion probabilistic models (DDPMs) in generative image modeling by learning a distribution of weights for the diffusion model, enabling the estimation of epistemic uncertainty through the variance across the model's predictions. Similarly, other approaches (Berry et al., 2024) train latent diffusion models (diffusion ensembles) for UQ by estimating the mutual information over a distribution of the models' weights, analogous to deep ensembles. However, these training-based UQ methods are challenging to implement, given that diffusion models often have billions of parameters, creating significant computation overhead

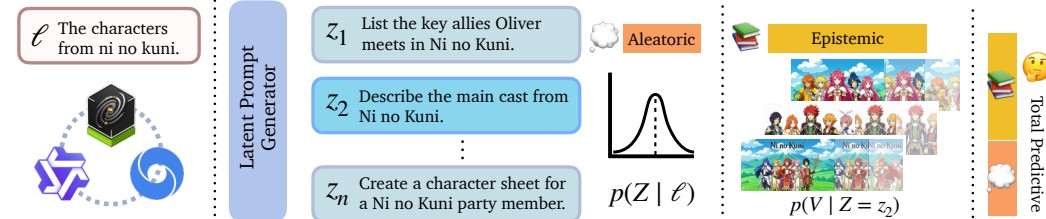

Figure 2: **S-QUBED architecture.** Given a text prompt $\ell$, our goal is to quantify the uncertainty of the video generation model. We first generate $n$ latent prompts consistent with $\ell$ in line with the prompt refinement used by video models, modeling the aleatoric uncertainty as the entropy of the distribution over latent prompts. Then, for each latent prompt, we generate $m$ videos, modeling the epistemic uncertainty as the conditional entropy of the distribution over generated videos. Finally, aggregating the two types of uncertainties yields the total predictive uncertainty.

during training or inference. Drawing insights from black-box UQ methods for LLMs (Manakul et al., 2023; Lin et al., 2023; Becker & Soatto, 2024) which utilize similarity-based techniques for efficient UQ, PUNC (Franchi et al., 2025) explores uncertainty quantification of generative image models in language space. By mapping generated images into language form using a VLM, PUNC leverages widely-used text-based similarity metrics (Zhang et al., 2019; Lin, 2004) to estimate epistemic and aleatoric uncertainty of text-to-image models. Although PUNC addresses the computation limitations of prior UQ methods for generative image models, PUNC is not applicable to video models. To our knowledge, this work is the first exploration of UQ for video world models.

## 3 PROBLEM FORMULATION

We examine uncertainty quantification of black-box text-conditioned video generation models, which map a text prompt $\ell \in \mathcal{O}$ to a video $v \in \mathcal{V}$ via an unknown stochastic model $f_\theta : \mathcal{T} \mapsto \mathcal{V}$ parametrized by weights $\theta$. Specifically, the video generation process is described by the model:

$$v \sim f_\theta(V \mid \ell), \tag{1}$$

where $v$ is sampled from the conditional distribution $f_\theta$. For an input prompt $v$, the video generation model has a measure of doubt (uncertainty) associated with the sampled video output $v$. This uncertainty arises from a variety of sources, e.g., vagueness in the conditioning input $\ell$, randomness in the physical evolution of the real-world, limited training data, etc. In this work, we are interested in quantifying the *total* predictive uncertainty associated with $v$, which can be broadly classified into two categories: *aleatoric* uncertainty and *epistemic* uncertainty.

## 4 UNCERTAINTY QUANTIFICATION OF GENERATIVE VIDEO MODELS

We present S-QUBED, an efficient method for uncertainty quantification of video generation models, summarized in Figure 2. Without loss of generality, we can decompose the video generation model in Equation (1) using a latent variable $z \in \mathcal{Z}$, modeling the video generation as a two-step process. In the first step, $z$ is sampled from the probability distribution $p(Z \mid \ell)$ conditioned on the input prompt $\ell$. In the second step, the video model samples the output video $v$ from the probability distribution $p(V \mid Z = z)$. Note that the application of latent variables is standard in generative modeling, e.g., in variational Bayesian learning (Kingma & Welling, 2013; Sohn et al., 2015; Bhattacharyya et al., 2019), enabling efficient learning and analysis of complex data-generation distributions. Consequently, we can rewrite Equation (1) in the form:

$$f_\theta(V \mid \ell) = \int_{z \in \mathcal{Z}} p(V \mid z, \ell) p(z \mid \ell) \mathrm{d}z = \int_{z \in \mathcal{Z}} p(V \mid z) p(z \mid \ell) \mathrm{d}z, \tag{2}$$

where we assumed conditional independence of $V$ and $\ell$, given $z$.

Note that the video generation model described by Equation (2) is not limiting. In fact, state-of-the-art text-to-video models refine a user's prompt using an LLM to generate a much more detailed prompt

that is passed into the video generation model. Hence, we can interpret Equation (2) as first sampling an instance of a fully-specified prompt $z$ from the conditional distribution defined by the input prompt $\ell$, e.g., given the input prompt "a cat doing something," $z$ may be the more specific prompt "a cat licking its paws before turning to the camera and meowing..." Subsequently, the video model generates the output video conditioned on $z$.

**Proposition 1** (Uncertainty Decomposition). *Define the total predictive uncertainty in the output video as the differential entropy $h(V \mid \ell)$ of the distribution $f_\theta(V \mid \ell)$. Then, this quantity can be decomposed as:*

$$h(V \mid \ell) = h(V \mid Z) + h(Z \mid \ell), \tag{3}$$

*where $h(V \mid Z)$ represents the epistemic uncertainty in $v$, and $h(Z \mid \ell)$ the aleatoric uncertainty.*

This is a standard decomposition. We provide the proof in Appendix B for completeness. In the rest of this section, we introduce our approach to estimating these components.

### 4.1 ALEATORIC UNCERTAINTY

Aleatoric uncertainty encompasses irreducible randomness from the vagueness (lack of sufficient specificity) of the conditioning inputs, e.g., "generate a video of a cat doing something." In video generation, vagueness in the input prompt increases the randomness of the conditional probability distribution $p(Z \mid \ell)$, which is represented by the second term $h(Z \mid \ell)$ in Equation (3). Note that $h(Z \mid \ell)$ is independent of $v$ since the source of uncertainty arises from the input prompt independent of the second stage of the video generation, e.g., the denoising process in video diffusion models. In particular, randomness in $Z$ cannot be reduced by training the video model on additional data under the assumption that we can model $p(Z \mid \ell)$ *almost* exactly.

As a measure of aleatoric uncertainty, we would expect $h(Z \mid \ell)$ to be positively correlated with the vagueness of the input prompt. For example, consider two input prompts: $\ell_1 =$ "a cat napping" and $\ell_2 =$ "a cat doing something". With $\ell_1$, the pdf of $p(Z \mid \ell_1)$ will be concentrated on the set:

$$\mathcal{A}(\ell_1) = \{\text{"a black cat napping", "a cat napping on a couch", "a cat snoring on a couch", \dots}\}. \tag{4}$$

However, with $\ell_2$, the pdf of $p(Z \mid \ell_2)$ will be concentrated on the set:

$$\mathcal{A}(\ell_2) = \{\text{"a black cat jumping", "a cat eating on a couch", "a cat meowing next to a door", \dots}\}. \tag{5}$$

Note that the elements of $\mathcal{A}(\ell_1)$ are more semantically-related (since $\ell_1$ is more specific) and are thus closer in the language (semantic) embedding space compared to elements in $\mathcal{A}(\ell_2)$. Hence, $p(Z \mid \ell_1)$ will have a lower entropy relative to $p(Z \mid \ell_2)$.

**Modeling the conditional latent distribution.** To compute $h(Z \mid \ell)$, we need to define a class of probability distributions that describe the latent-generation process. In this work, we model $p(Z \mid \ell)$ in a language embedding space using the Von-Mises Fisher (VMF) distribution (Fisher, 1953; Jupp & Mardia, 1989), drawing insights from prior work (Robertson, 2004; Banerjee et al., 2005; Gopal & Yang, 2014).

The Von-Mises Fisher (VMF) distribution describes a $n$-dimensional probability distribution on the $(n-1)$-sphere over unit vectors embedded in $\mathbb{R}^n$, with the probability density function (pdf):

$$f_n(x, \mu, \kappa) = C_n(\kappa) \exp(\kappa \mu^\mathsf{T} x), \tag{6}$$

with parameters $\mu$ and $\kappa$ denoting the mean direction and concentration parameters, where:

$$C_n(\kappa) = \frac{\kappa^{n/2-1}}{(2\pi)^{n/2} I_{n/2-1}(\kappa)}, \tag{7}$$

with $I_{n/2-1}$ representing the modified Bessel function of the first kind. The concentration parameter functions analogously to the inverse variance, providing a measure of the spread of the distribution.

We need samples from $p(Z \mid \ell)$ to fit the VMF distribution. Collecting such data is typically prohibitively expensive. To overcome this challenge, we leverage LLMs as cost-effective generative models of $p(Z \mid \ell)$, noting that video models generally use LLMs to refine prompts prior to generating videos.

Specifically, given an input prompt $\ell$, we generate $N$ *compatible*-but-more-specific prompts from an LLM. A generated prompt is *compatible* with the input prompt if the generated prompt is consistent with, i.e., *entails*, the input prompt. However, the converse need not be true: the input prompt might be underspecified. Subsequently, we compute language embeddings from an embedding model, e.g., SentenceFormer (Reimers & Gurevych, 2019). Although we could directly fit a VMF to the language embeddings, we project the language embeddings to a lower-dimensional subspace $\mathbb{R}^n$ using principal component analysis (PCA) to avoid numerical instability associated with high-dimensional spaces. We estimate the parameters $\mu$ and $\kappa$ of the VMF distribution in closed-form using approximate methods (Jupp & Mardia, 1989; Sra, 2012), circumventing iterative optimization methods.

**Estimating Aleatoric Uncertainty.** Given $p(Z \mid \ell)$, we can compute the aleatoric uncertainty $h(Z \mid \ell)$ of $v$ in closed-form via:

$$h(Z \mid \ell) = -\log(C_n(\kappa)) - \frac{\kappa}{\mu_{z|\ell}}\mathbb{E}_Z[Z \mid \ell](\kappa), \tag{8}$$

where $Z \sim \mathrm{VMF}(\mu, \kappa)$ and $C_n$ represents the normalization constant given by Equation (7). The expected value of the VMF is given by $\mathbb{E}_Z[Z \mid \ell](\kappa) = W_n(\kappa)\mu_{z|\ell}$, where $W_n = \frac{I_{n/2}(\kappa)}{I_{n/2-1}(\kappa)}$ with the modified Bessel function of the first kind $I_{n/2}$. We summarize the method for computing aleatoric uncertainty in Algorithm 1.

---

**Algorithm 1:** S-QUBED: Aleatoric Uncertainty Quantification of Generative Video Models

---

**AleatoricUncertainty** $(f, \ell)$:

    **Input:** Video Model $f$, Input Prompt $\ell$;

    **Output:** Aleatoric Uncertainty $h(Z \mid \ell)$;

    $\mathcal{A}(\ell) \leftarrow \mathrm{Embed}(\mathrm{LLM}(\ell))$ ;           // Construct $\mathcal{A}(\ell)$ from an LLM/VLM

    $\mu_{z|\ell}, \kappa_{z|\ell} \leftarrow \mathrm{VFit}(\mathcal{A}(\ell))$ ;           // Estimate $p(Z \mid \ell)$ with a VMF

    $h(Z \mid \ell) \leftarrow Equation$ (8) ;      // Compute aleatoric uncertainty $h(Z \mid \ell)$

    **return** $h(Z \mid \ell)$;

---

## 4.2 Epistemic Uncertainty

Epistemic uncertainty represents the measure of doubt associated with a lack of knowledge, which generally results from insufficient training data (e.g., Figure 1). As a result, epistemic uncertainty is *reducible* by providing additional training data to the model. In Equation (3), $h(V \mid Z)$ represents the epistemic uncertainty of the generated video $v$, where the uncertainty arises from the limited knowledge of the video model about concepts expressed by the latent variable $z \in \mathcal{Z}$.

For example, consider a video model trained entirely on internet videos of cats and dogs performing different activities, e.g., running, eating, jumping, meowing/barking. Now, when asked to generate a video of "a lion roaring in the wild", the video model might generate different videos across different runs, with some showing a large cat meowing in a park with significant tree canopy, others showing a cat making *barking-like* sounds in a forest, etc. Although the generated videos are all conditioned on semantically-consistent latent variables, the generated videos might be semantically-inconsistent, since the video model has not been trained on videos of lions. This uncertainty in the generated videos can be described as *epistemic* and is captured by the entropy term $h(V \mid Z)$.

**Estimating Epistemic Uncertainty.** Note that we can express $h(V \mid Z)$ in the form:

$$h(V \mid Z) = \mathbb{E}_{z \sim p(z|\ell)}[h(V \mid Z = z)], \tag{9}$$

which can be interpreted as the expected entropy of the distribution of generated videos conditioned on sampled latent states $z$ from the conditional distribution $p(z \mid \ell)$. Computing $h(V \mid Z)$ is challenging for two reasons: (i) we do not have an explicit model of $p(V \mid Z = z)$ which is required to compute $h(V \mid Z = z)$, and (ii) even with an analytical expression for $p(V \mid Z = z)$, computing $h(V \mid Z)$ would require evaluating a double integral, which is intractable in general.

To address the first challenge, we approximate the conditional distribution $p(V \mid Z = z)$ using a VMF distribution with the parameters $\mu$ and $\kappa$ estimated from samples drawn from the video model.

Likewise, we approximate the expectation in Equation (9) using Monte-Carlo sampling to address the second challenge, which we describe in greater detail.

First, we sample a set of latent variables $\mathcal{E}_{z|\ell}$ conditioned on the input prompt $\ell$ from the distribution $p(Z \mid \ell)$, with each $z \in \mathcal{E}_{z|\ell}$ representing specific instances of prompts entailing the input prompt. For each $z$, we estimate the distribution $p(V \mid Z = z)$ by generating a set of videos $\mathcal{E}_{v|z}$ from the video model, conditioned on $z$. Subsequently, we embed these videos with a video embedding model, e.g., S3D (Miech et al., 2020) and fit a VMF to the samples in $\mathcal{E}_{v|z}$. Afterwards, we compute the entropy $h(V \mid Z = z)$ with:

$$h(V \mid z) = -\log(C_n(\kappa_{v|z})) - \frac{\kappa_{v|z}}{\mu_{v|z}}\mathbb{E}_{v|z}[V \mid Z = z](\kappa_{v|z}), \tag{10}$$

using the estimated VMF parameters $\mu_{v|z}$ and $\kappa_{v|z}$. Finally, we compute an empirical estimate of the expectation of $h(V \mid Z = z)$ over z sampled from $p(Z \mid \ell)$. We outline these steps in Algorithm 2.

---

**Algorithm 2:** S-QUBED: Epistemic Uncertainty Quantification of Generative Video Models

---

**EpistemicUncertainty** $(f, \ell)$**:**

    **Input:** Video Model $f$, Input Prompt $\ell$;

    **Output:** Epistemic Uncertainty $h(V \mid z)$;

    $\mathcal{E}_{z|\ell} \leftarrow \text{Embed}(\text{LLM}(\ell))$ ;            // Construct $\mathcal{E}_{z|\ell}$ from an LLM/VLM

    **foreach** $z \in \mathcal{E}_{z|\ell}$ **do**

        $\mathcal{E}_{v|z} \leftarrow \text{Embed}(f(V \mid z))$ ;         // Construct $\mathcal{E}_{v|z}$ from $f$

        $\mu_{v|z}, \kappa_{v|z} \leftarrow \text{VFit}(\mathcal{E}_{v|z})$ ;       // Estimate $p(V \mid Z = z)$ from $f$

        $h(V \mid Z = z) \leftarrow Equation$ (10) ;      // Compute entropy $h(V \mid Z = z)$

    **end**

    $h(V \mid Z) \leftarrow Equation$ (9) ;    // Compute epistemic uncertainty $h(V \mid Z)$

    **return** $h(V \mid Z)$;

---

## 5 Experiments

We examine the effectiveness of S-QUBED in uncertainty quantification of generative video models, specifically exploring the following questions: (i) *How do we evaluate uncertainty calibration of video models?* (ii) *Are the total predictive uncertainty estimates computed by S-QUBED calibrated?* (iii) *Can S-QUBED effectively estimate both aleatoric and epistemic uncertainty?*

### 5.1 Evaluation Setup

We describe the datasets, models, and metrics used in evaluating our proposed method.

**Datasets.** We evaluate S-QUBED on two large-scale video generation datasets, VidGen-1M (Tan et al., 2024) and Panda-70M (Chen et al., 2024). Using GPT-5-nano (OpenAI, 2025), we classify the videos in each dataset into five broad categories: animals, food, games, people, and other, a standard approach with video datasets. From each dataset, we subsample about 200 video generation tasks uniformly from each category for evaluation. To address issues with missing video data/metadata in some of the datasets, we sample additional videos from other categories, minimally changing the uniformity of the evaluation dataset.

**Implementation.** We evaluate S-QUBED on the Cosmos-Predict2 video model (NVIDIA Cosmos, 2025) using the official implementation, which utilizes a text-to-image-to-video pipeline for text conditioning that generates an image from a text prompt, which is used as input to an image-to-video model. We implement our proposed method by sampling 10 latent states, $z_{1:10} \sim p(Z|\ell)$, and subsequently 10 generated videos per latent state, $v_{1:10}^i \sim p(v_j^i|Z = z_i)$. We explore alternative generative video models, including CogVideoX (Yang et al., 2024), Veo 3 (DeepMind, 2025), and OpenSora (Peng et al., 2025) in Appendix C, covering a wide range of open-source and closed-source models. However, due to practical limitations on the number of permissible generation requests or prohibitive compute cost, we explore different hyperparameters to enable effective implementations with generation or cost constraints.

## 5.2 HOW DO WE EVALUATE UNCERTAINTY CALIBRATION OF VIDEO MODELS?

Uncertainty calibration of video generation models has been underexplored, evidenced by the lack of purpose-specific calibration metrics. Widely-used calibration metrics, such as the expected calibration error (ECE) and maximum calibration metrics (MCE) apply only to evaluation settings with discrete ground-truth answers and errors, e.g., with multiple-choice questions, making them unsuitable in video generation tasks with real-valued task errors. Consequently, we propose appropriate metrics for evaluating the calibration of the uncertainty estimates of video models. Specifically, we examine the Kendall rank correlation (Kendall's $\tau$) (Kendall, 1938) between the video model's uncertainty estimates and an applicable accuracy metric, which captures the degree of monotonicity between uncertainty and accuracy. We do not utilize Pearson's rank correlation (Galton, 1895) due to its assumptions of linearity and normally-distributed data and likewise do not use the Spearman's rank correlation coefficient (Spearman, 1987) due to its high sensitivity to outliers.

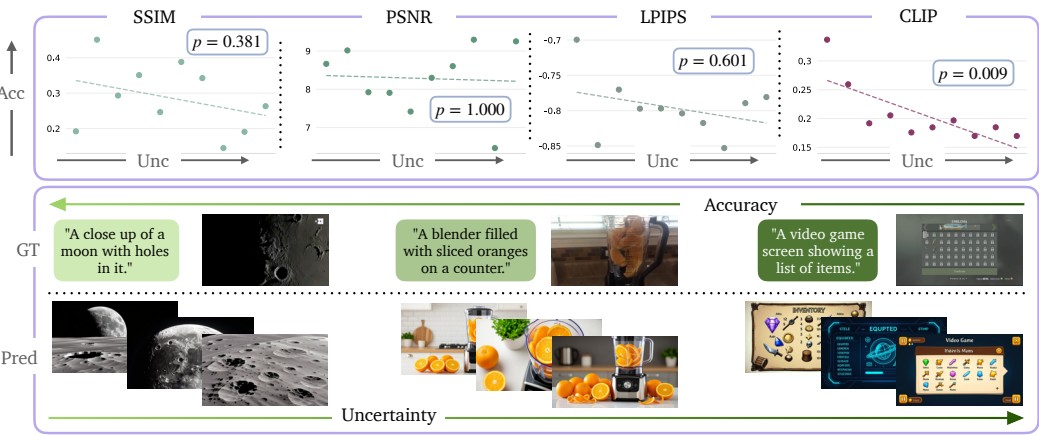

Figure 3: **Calibration Metrics for Video Models.** *Top*: We examine the statistical significance of the Kendall rank correlation between uncertainty and widely-used perceptual metrics. We find that the CLIP cosine similarity score provides the most significant correlation. *Bottom*: With the CLIP accuracy metric, we observe that low human-annotated uncertainty corresponds to smaller variance in the generated videos and greater accuracy with respect to the ground-truth video. As uncertainty increases, video prediction accuracy decreases.

To compute the rank correlation coefficient, we use the SSIM, PSNR, LPIPS, and CLIP score metrics. To identify the best metric for assessing calibration, we select 10 generation tasks from the Panda-70M datasets and rank the tasks in order of increasing uncertainty based on the vagueness of the text prompt for the task. Note that the vagueness in the prompt directly corresponds to aleatoric uncertainty, making it an effective proxy measure. Given the human-annotated rankings, we compute the Kendall rank correlation between uncertainty and each accuracy metric along with a $p$-value, which provides a measure of the statistical significance of the correlation. While Panda-70M dataset consists of tasks with a broad range of descriptive detail from vague to very specific, VidGen-1M consists of relatively well-detailed tasks. As a result, we do not sample from VidGen-1M, given the less observable variation in the aleatoric uncertainty. We sample the tasks from Panda-70M dataset to retain the distribution of instruction detail.

We summarize our results in Figure 3. In Panda-70M, the CLIP score metric is strongly negatively correlated with uncertainty at the 99% significance level. In contrast, the other perceptual metrics lack a statistically significant correlation with uncertainty. This finding is not entirely surprising, since CLIP captures semantic information that better reflects the accuracy of the generation task, unlike the other perceptual metrics which are more susceptible to differences in visual changes.

Moreover, we visualize the text prompt, ground-truth video, and the first frame of the generated videos for a few tasks in Figure 3, ranging from low to high uncertainty (rank). We observe that when uncertainty is low, the model tends to generate very similar videos, which are also close to the ground-truth, resulting in high accuracy with respect to the CLIP score. As we vary the uncertainty of the model, we observe greater variance in the generated videos accompanied by notably lower

CLIP scores (compared to the other metrics), further demonstrating the utility of the CLIP score as an accuracy metric.

## 5.3 ARE OUR UNCERTAINTY ESTIMATES CALIBRATED?

We examine the calibration of our uncertainty estimates in VidGen-1M and Panda-70M, using the CLIP score accuracy metric given its effectiveness in assessing calibration. We first compute the total predictive uncertainty associated with each video task using S-QUBED, and then evaluate the Kendall rank correlation. We define the accuracy of each task as the mean CLIP score across all generated videos for that task.

Figure 4 (left) presents results for Panda-70M. We observe a statistically significant negative correlation (99% confidence level) between the total uncertainty computed using S-QUBED and the CLIP score, demonstrating calibration of the uncertainty estimates. The results highlight that as the uncertainty of the video model decreases, its accuracy increases. Likewise, in VidGen-1M, the total predictive uncertainty is negatively correlated with the CLIP score at the 89.9% confidence level. From Figure 4, we see that when the total predictive uncertainty estimates is small ("A"), the video model generates more accurate videos; in contrast, in tasks with high estimated uncertainty ("B"), the video model is less accurate.

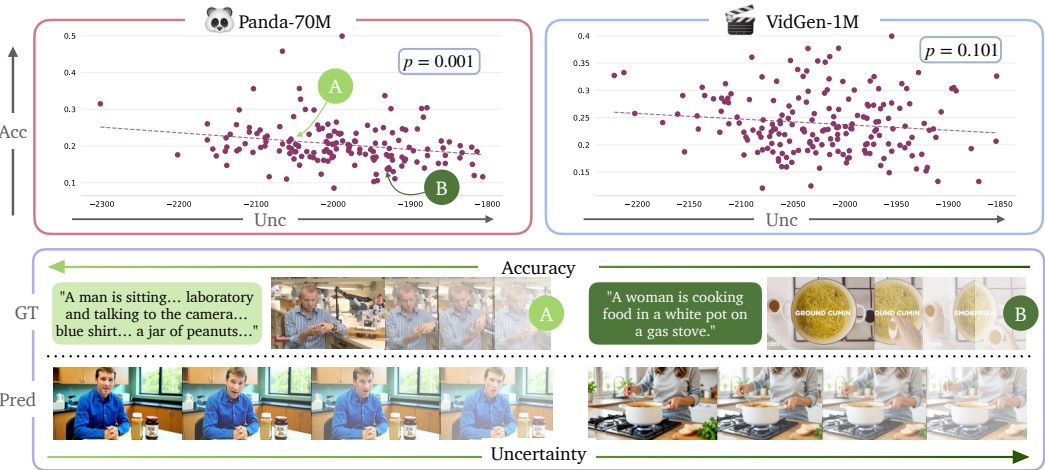

Figure 4: **Total Predictive Uncertainty for Video Models.** We assess the calibration of the total predictive uncertainty computed by S-QUBED. *Top*: correlation between video prediction accuracy and total uncertainty for Panda-70M and VidGen-1M . We observe a statistically significant correlation between accuracy and uncertainty for both datasets, signified by the small *p*-values. *Bottom*: visualization of two samples from Panda-70M.

## 5.4 CAN S-QUBED EFFECTIVELY ESTIMATE BOTH ALEATORIC AND EPISTEMIC UNCERTAINTY?

We examine the performance of S-QUBED in decomposing total uncertainty into aleatoric and epistemic uncertainty. To effectively assess calibration of aleatoric uncertainty, we consider a subset of each dataset where the epistemic uncertainty is almost zero and compute the rank correlation between the aleatoric uncertainty of these samples and the CLIP score. Likewise, to evaluate calibration of epistemic uncertainty, we compute the rank correlation between the epistemic uncertainty and the CLIP score for samples with relatively zero aleatoric uncertainty. In practice, we select samples with the lowest aleatoric or epistemic uncertainty, accordingly.

In Figure 5, we visualize the Kendall rank correlation between the aleatoric and epistemic uncertainty and the CLIP score in both datasets. In Panda-70M, we find that aleatoric and epistemic uncertainty are negatively correlated with accuracy at the 94.5% and 98.3% confidence level. Similarly, in VidGen-1M, we observe a statistically significant negative correlation between aleatoric and epistemic

uncertainty and the accuracy at the 92.3% and 91.7%, respectively. These results highlight that S-QUBED can decompose total uncertainty effectively into its aleatoric and epistemic components

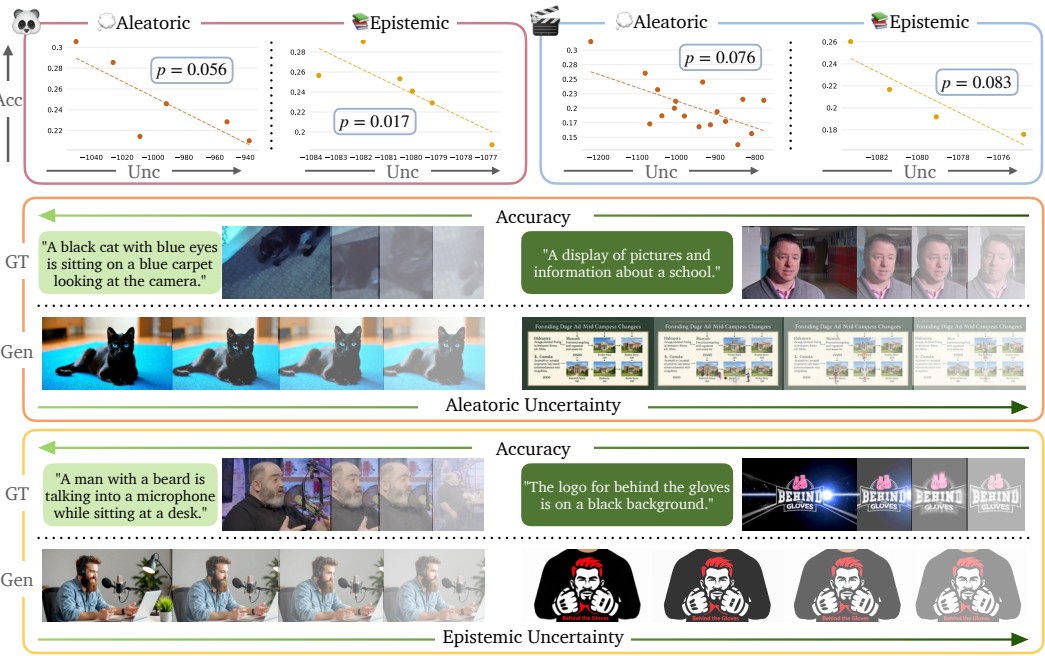

Figure 5: **Disentangling Aleatoric and Epistemic Uncertainty for Video Models.** We demonstrate the calibration of the aleatoric uncertainty estimates of S-QUBED in tasks with no epistemic uncertainty, showing statistically significant negative correlation. We do the same for epistemic uncertainty.

Further, we visualize text prompts, ground-truth-videos, and generated videos in tasks with low and high estimated aleatoric uncertainty. We observe that in the low-uncertainty case, the video model achieves high accuracy, unlike the high-uncertainty case, where the prediction accuracy is significantly lower. Similarly, we provide some visualizations in the case with low and high estimated epistemic uncertainty, showing the negative correlation between S-QUBED's estimated epistemic uncertainty and video prediction accuracy. Notably, the model does not know the specific "Behind the Gloves" logo and thus generates a generic logo, unlike predicting the person in the human-centric videos.

## 6 CONCLUSION

We present a framework for empowering video models to express their uncertainty, a critical capability for safety. Concretely, we introduce a metric for measuring the calibration of UQ methods for video models and present a calibrated UQ method for video models. Our methods utilizes latent modeling to estimate both aleatoric and epistemic uncertainty, without making any limiting assumptions. Further, we provide an open-source video dataset for benchmarking UQ methods for video models. Our experiments demonstrate the calibration of our proposed method and its effectiveness in disentangling aleatoric and epistemic uncertainty.

## 7 LIMITATIONS AND FUTURE WORK

S-QUBED requires generating multiple videos from the video model to estimate epistemic uncertainty, which poses some computational overhead. Future work will explore more efficient strategies for sampling videos from the video model, e.g., in the latent space of the video model. Beyond the two benchmark datasets considered in this work, we will explore extensions to new datasets to augment the UQ dataset curated for benchmarking calibration. In addition, future work will examine the application of our method to new open-source models, as they become available.

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
