## A  EVALUATION SETUP

We provide additional details on the evaluation setup.

**Metrics.** We consider the following standard video accuracy metrics: structural similarity index measure (SSIM) (Wang et al., 2004), peak signal-to-noise ratio (PSNR), learned perceptual image patch similarity (LPIPS) (Zhang et al., 2018b), and CLIP cosine similarity score. Note that the SSIM, PSNR, and LPIPS primarily assess visual fidelity while the CLIP score captures more semantic information. We take the negative of the LPIPS score to transform it from an error metric to an accuracy metric. To compute the perceptual metrics, we resize all videos spatially to the same dimensions and subsample the longer videos to ensure that all videos have the same duration. For CLIP, we map both the ground-truth video $v^{\text{gt}}$ and all the generated videos $v_j^i$ to the visual-semantic space using CLIP. We compute the mean of each metric over all generated videos per task, which represents the assigned value of the metric for that task. Prior work Wan et al. (2025) on video evaluation has demonstrated the effectiveness of CLIP in capturing both visual and semantic information, motivating its inclusion in our study.

## B  PROOFS

**Proposition 1** (Uncertainty Decomposition)**.** *Define the total predictive uncertainty in the output video as the differential entropy $h(V \mid \ell)$ of the distribution $f_\theta(V \mid \ell)$. Then, this quantity can be decomposed as:*

$$h(V \mid \ell) = h(V \mid Z) + h(Z \mid \ell), \tag{3}$$

*where $h(V \mid Z)$ represents the epistemic uncertainty in $v$, and $h(Z \mid \ell)$ the aleatoric uncertainty.*

*Proof.* The entropy of a random variable quantifies its associated uncertainty. Given the probability distribution $f_\theta(V \mid \ell)$, we find its entropy by:

$$h(V \mid \ell) = -\int_{v \in \mathcal{V}} f_\theta(V \mid \ell) \log(f_\theta(V \mid \ell)) \, \mathrm{d}v \tag{11}$$

$$= -\int_{v \in \mathcal{V}} \int_{z \in \mathcal{Z}} p(V \mid z) p(z \mid \ell) \log(p(V \mid z) p(z \mid \ell)) \, \mathrm{d}z \, \mathrm{d}v, \tag{12}$$

where we incorporate the latent state generation step introduced in Equation (2). We can then decompose the $\log$ terms into two components:

$$h(V \mid \ell) = -\int_{v \in \mathcal{V}} \int_{z \in \mathcal{Z}} p(V \mid z) p(z \mid \ell) \left( \log(p(V \mid z)) + \log(p(z \mid \ell)) \right) \, \mathrm{d}z \, \mathrm{d}v \tag{13}$$

$$= -\left( \int_{z \in \mathcal{Z}} p(z \mid \ell) \int_{v \in \mathcal{V}} p(V \mid z) \log(p(V \mid z)) \mathrm{d}v \mathrm{d}z \right)$$

$$- \left( \int_{z \in \mathcal{Z}} \left( \int_{v \in \mathcal{V}} p(V \mid z) \mathrm{d}v \right) p(z \mid \ell) \log(p(z \mid \ell)) \mathrm{d}z \right), \tag{14}$$

where Equation (14) applies the Fubini-Tonelli theorem. We note that each term of Equation (14) is an entropy itself:

$$h(V \mid \ell) = -\left( \int_{z \in \mathcal{Z}} p(z \mid \ell) h(V \mid Z = z) \mathrm{d}z \right) - \left( \int_{z \in \mathcal{Z}} p(z \mid \ell) \log(p(z \mid \ell)) \mathrm{d}z \right) \tag{15}$$

$$= h(V \mid Z) + h(Z \mid \ell). \tag{16}$$

We recognize that the first term $h(V \mid Z)$ eliminates uncertainty in prompt ambiguity, and thus signifies the epistemic uncertainty in video generation. On the other hand, the second term $h(Z \mid \ell)$ is independent of the video model, but rather only depends on the vagueness of the input prompt, signifying aleatoric uncertainty. □

# C    ADDITIONAL EXPERIMENTS

We explore the effectiveness of S-QUBED across closed-source and open-source video generation models, examining its robustness to different hyperparameters, e.g., the number of latent prompts per task and the number of generated videos per latent prompt. In particular, we consider the following state-of-the-art video models: CogVideoX Yang et al. (2024), Veo 3 DeepMind (2025), and OpenSora Peng et al. (2025). We note that these models impose different constraints that introduce practical limitations on the implementation of our method. Specifically, the closed-source model Veo 3 places a stringent cap on the number of generation tasks that a user can submit, limiting the number of latent prompts and latent videos that can be used in S-QUBED. Moreover, Veo 3 is expensive to run, introducing additional challenges. Likewise, OpenSora Peng et al. (2025) requires significant compute infrastructure with large GPU memory for video generation, e.g., an A100/H100 GPU, a potential practical bottleneck. Here, we examine if these challenges could be overcome through a more compatible choice of hyperparameters without compromising the calibration of S-QUBED. We present the additional results in the order of decreasing amount of latent prompts and latent videos. To assess calibration, we compute the correlation between the predicted uncertainty and the observed accuracy of the video model, noting that stronger negative correlation indicates better calibration of the video model. In these experiments, we use biweight midcorrelation as the correlation measure due to its robustness to outliers compared to other measures of correlation Song et al. (2012), which is especially important when analyzing data containing only a few data points. Further, we evaluate calibration on the Panda-70M dataset, given its greater diversity in the task descriptions.

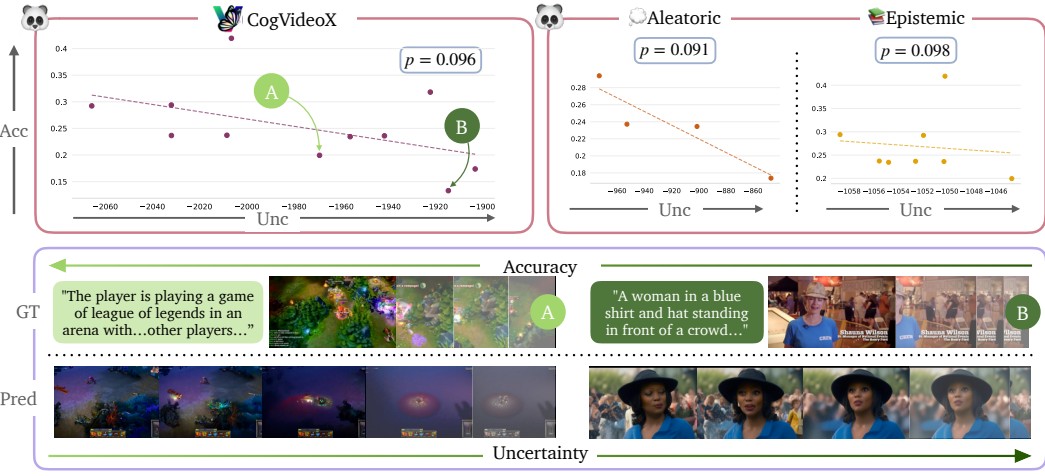

Figure 6: **Uncertainty Quantification of CogVideoX.** *Top:* S-QUBED computes calibrated uncertainty estimates for CogVideoX, decomposing the total uncertainty into trustworthy aleatoric and epistemic uncertainty components. *Bottom:* We demonstrate alignment of the uncertainty estimates with video accuracy, showing that when the video model is more uncertain, the generated video is likely to be more inaccurate, as visualized by Tasks A and B.

## C.1    RESULTS WITH COGVIDEOX

We quantify the uncertainty of CogVideoX in a random subset of 10 tasks in the Panda-70M dataset. For each task, we generate 10 latent prompts and 10 videos per latent prompt. From the resulting videos, we compute the total predictive uncertainty of the model, decomposed into its aleatoric and epistemic components. Figure 6 summarizes the quantitative and qualitative calibration results. First, as visualized in the top-left plot, we observe that the total predictive uncertainty estimates are well-calibrated. Specifically, the estimated uncertainty of the video model is negatively correlated with its accuracy, with a correlation coefficient of $-0.527$ at the 90% significance level. The aleatoric and epistemic estimates are also negatively correlated with accuracy at the 90% significance level, with coefficients of $-0.909$ and $-0.625$, respectively. These findings underscore that the video model's predicted uncertainty is well-aligned with the observed accuracy of the model. In other

words, when the video model is uncertain about its generated video, the synthesized video is more likely to be inaccurate. In essence, S-QUBED's uncertainty estimates are trustworthy. Further, we observe that the uncertainty estimates are well-aligned with human intuition, as demonstrated by the two video generation tasks shown in Figure 6 where S-QUBED predicts that Task A has a higher uncertainty compared to Task B. From the ground-truth and generated videos in the figure, we see that these uncertainty estimates are consistent with the relative accuracy of the video model on both tasks.

## C.2 RESULTS WITH VEO 3

We examine the amenability of S-QUBED to closed-source models with significant cost overhead. For example, the strict API limits on the maximum number of videos generated by Veo 3 presents notable challenges to uncertainty quantification. Likewise, each video generation task is also expensive to run. To address these challenges, we examine the calibration of S-QUBED when *fewer* latent prompts are used with *fewer* generated videos per latent prompt. Concretely, we consider five tasks, where we generate five latent prompts for each video generation task and four videos per latent prompt to estimate the uncertainty of Veo 3. We provide the calibration results in Figure 7. We find that the uncertainty estimates are strongly negatively correlated with accuracy at the $90\%$ significance level with a correlation coefficient of $-0.838$. Likewise, the aleatoric uncertainty estimates are well-calibrated with a correlation coefficient of $-0.937$. We do not observe a statistically significant correlation between the estimated epistemic uncertainty and the accuracy of the video model, which is not surprising given that the correlation is being computed using only a few video generation tasks, making the computed correlation coefficient more susceptible to the effects of outliers. Qualitatively, as noted in our earlier discussion, the video model's uncertainty estimates are interpretable. As shown in Figure 7, Veo 3 is more uncertain about generating characters from Ni no Kuni compared to generating a video showing the Boston Celtics playing a basketball game. We see that the estimated uncertainty is well-aligned with the accuracy of the generated videos.

Not significant

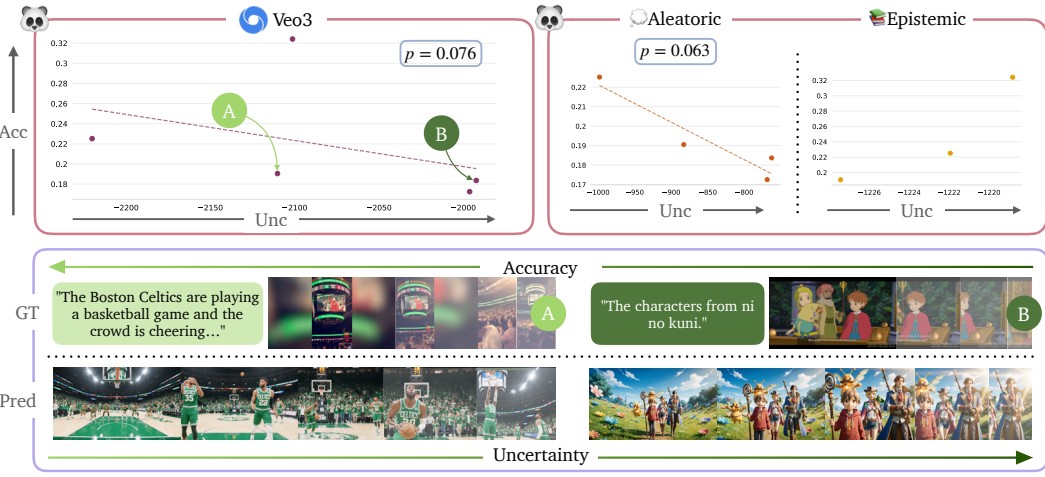

Figure 7: **Uncertainty Quantification of Veo 3.** *Top:* S-QUBED enables Veo 3 to express calibrated uncertainty estimates, although stringent limits on the number of generated videos could hinder the observation of statistically significant levels of correlation between epistemic uncertainty and accuracy. *Bottom:* The estimated uncertainty estimates are well-aligned with the accuracy of the generated videos. For example, the generated video and uncertainty estimate suggest that Veo 3 does not know the characters from Ni no Kuni compared to its knowledge of the Boston Celtics.

## C.3 RESULTS WITH OPENSORA

We examine the applicability of S-QUBED to open-source models that require significant compute resources, e.g., models requiring large GPUs such as the A100/H100. Here, we evaluate the calibration of S-QUBED with OpenSora as a representative video model, running on an A100 GPU with 80GB memory. We estimate the uncertainty of OpenSora on 15 tasks from Panda-70M, generating three

latent prompts and three videos per latent prompt. Figure 8 shows the calibration results, highlighting that the estimated uncertainty is negatively correlated with accuracy at a $90\%$ significance level with a coefficient of $-0.481$. Further, we find that S-QUBED produces well-calibrated aleatoric uncertainty estimates; however, similar to the experiments with Veo 3, we do not observe a statistically significant negative correlation between the epistemic uncertainty estimates and accuracy, likely due to the lack of sufficient generated videos to mitigate the effect of outliers. Nonetheless, the results underscore that S-QUBED produces well-calibrated total predictive uncertainty estimates, which is further demonstrated by the ground-truth and generated videos in Figure 8. The generated video in Task A is more accurate compared to the generated video in Task B, in line with the video model's uncertainty estimates.

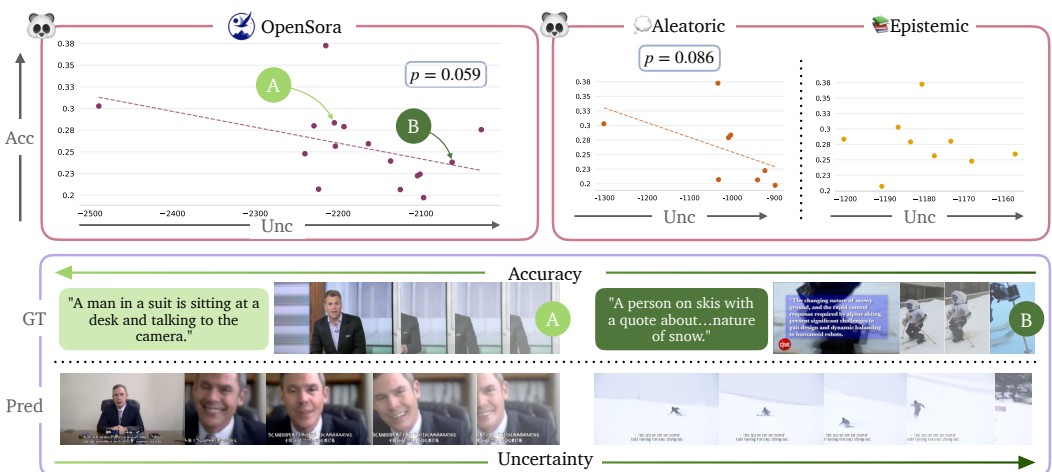

Figure 8: **Uncertainty Quantification of OpenSora.** *Top:* Empowered by S-QUBED, OpenSora produces uncertainty estimates that are well-aligned with accuracy. Unlike the aleatoric uncertainty estomates, the epistemic uncertainty does not show a statistically significant negative correlation since only a few generated videos are used in estimating the correlation. *Bottom:* The uncertainty estimates are trustworthy, with larger uncertainty estimates corresponding to greater errors between the ground-truth and generated videos.