# OpenReview forum: "How Confident are Video Models? Empowering Video Models to Express their Uncertainty"
_ICLR.cc/2026/Conference — Submitted to ICLR 2026_

### Official Review · Reviewer_Mt95 · 2025-10-19

**Soundness:** 1
**Presentation:** 2
**Contribution:** 2
**Rating:** 2
**Confidence:** 5

**Summary:**

Text-to-video models are quickly improving and creating excitement both among researchers and users of AI.  However, like LLMs, these models are prone to hallucinate details of their output, especially when the input prompt is underspecified or underrepresented in training data.  To address this challenge, this work presents the first (to their knowledge) study of uncertainty quantification in text-to-video models.  They propose a black-box UQ method based on the epistemic/aleatoric decomposition to help identify when a text-to-video model is likely to hallucinate, and also plan to release a dataset of 40K videos for benchmarking UQ.

**Strengths:**

Effective uncertainty quantification is a central pillar in creating trustworthy AI systems.  While most focus on UQ in deep learning has been in image classification and more recently LLMs, it is important that these tools are extended to other fields and application areas, for example robotics or other generative media besides text.  This paper aims to take the first step towards developing a framework and tools for UQ in text-to-video systems.  This is a very solid motivation, and creates the potential for a significant contribution.

**Weaknesses:**

The main weakness I find is that this paper does not carefully treat the concepts of epistemic and aleatoric uncertainty, in particular by treating them primarily in terms of the input prompt rather than as properties that depend on the interaction between the model, its capacity, and the data distribution. Aleatoric uncertainty is described as randomness from prompt vagueness, while epistemic uncertainty is tied to a lack of model knowledge. This framing assumes these uncertainties are intrinsic to the prompt, but in practice, they are model- and data-dependent. For instance, if the entire training set consists of videos of cats napping on purple beds in the backs of pickup trucks, then the prompt “a cat napping on a purple bed in the back of a pickup truck” would still display high aleatoric uncertainty, not because the prompt lacks specificity, but because the data distribution itself is highly variable in that region. By focusing almost entirely on prompt semantics, the paper overlooks the fact that the distinction between epistemic and aleatoric uncertainty depends fundamentally on the model and the data it has seen.

This conceptual problem extends directly into the method. The decomposition in Equation (3) is presented as a principled separation between epistemic and aleatoric uncertainty, but in practice both quantities depend on the behavior and biases of the specific models used to estimate them. The authors estimate aleatoric uncertainty by prompting an LLM to generate refined textual variants and epistemic by sampling multiple videos from the same generative model. Both steps produce variability that arises from model architectures and training data of the various models, not from isolated intrinsic uncertainty types. What they call aleatoric uncertainty reflects the LLM’s own distribution, while their epistemic uncertainty reflects the video embedding model’s representation space, making the split depend on implementation choices rather than underlying epistemic principles. As a result, the decomposition is not theoretically or empirically meaningful.

Beyond these conceptual issues, the method relies on untested and implausible assumptions. The independence assumption discards dependence between the text prompt and the generated video, which is unlikely to hold in text-to-video generation. The estimation of entropy in embedding spaces further introduces arbitrary geometric distortions, since the embedding dimensions and projection have a major effect on the computed entropies. The authors provide no sensitivity analysis or justification for these choices, leaving the reported uncertainty values largely uninterpretable.

The experimental evaluation generally lacks rigor. The decision to use CLIPScore as the primary accuracy metric is based on a small 10 sample correlation study, an inadequate basis for methodological justification. The subsequent experiments that claim to disentangle aleatoric and epistemic uncertainty depend on opaque subsetting of data where one component is deemed zero according to the authors’ own estimators, introducing circular reasoning. These experimental protocols make the reported calibration and correlation results difficult to trust.

Overall, the proposed decomposition lacks solid conceptual grounding, the implementation does not meaningfully separate uncertainty types, and the empirical evaluation does not convincingly support the claims.

**Questions:**

See weaknesses.

---

> ### Author Response · Authors · 2025-11-23
> **Response to Reviewer Mt95 (Part I)**
>
> Thank you for recognizing the timeliness of our work. We are
> grateful for your valuable feedback, which we found to be really
> helpful and constructive. We group our response into the following
> sections:
>
> **Clarification on Types of Uncertainties.** We apologize for the
> misunderstanding in the difference between aleatoric and epistemic
> uncertainty and the connection between the specificity of a prompt
> and the associated aleatoric uncertainty. We clarify these notions in
> our response. From [1, cf. Page 2], aleatoric uncertainty refers to
> variability or doubt due to inherently random effects. In contrast,
> epistemic uncertainty refers to doubt due to a lack of knowledge. Im-
> portantly, total uncertainty can be decomposed into two components:
> (1) a irreducible part which is the aleatoric uncertainty, and (2) a
> reducible part which is the epistemic uncertainty. This definition
> of uncertainty from [1, cf. Page 2] is consistent with the definition
> provided in our paper.
> We emphasize that for a given task, aleatoric uncertainty is
> irreducible. In other words, we **cannot** reduce aleatoric uncertainty,
> even if we collect an **infinite** amount of data. Likewise, no model
> can reduce the aleatoric uncertainty, precisely because there is no
> interaction between the model, capacity, or data that influences
> aleatoric uncertainty. In text-to-video generation, aleatoric uncertainty
> is an inherent property of the task, which is defined by the prompt,
> highlighting the connection between aleatoric uncertainty and the
> vagueness of the prompt.
> It appears that the notion of epistemic uncertainty might have
> been conflated with aleatoric uncertainty. Unlike aleatoric uncertainty,
> epistemic uncertainty can be reduced by gaining more knowledge,
> e.g., with more training data or a larger, more expressive model.
> We reiterate that these notions of uncertainty are consistent with
> standard formal definitions.
> Now, applying these definitions to the example you described, we
> observe that the task “a cat napping on a purple bed in the back of
> a pickup truck” always has some aleatoric uncertainty. For example,
> we do not know what time of the day it is, what the weather looks
> like, what the cat looks like, etc. Regardless of how large our training
> dataset is, this uncertainty cannot be reduced—it is captured by the
> vagueness in the prompt. On the other hand, consider if we initially
> used a small training dataset containing only dogs, the video model
> may not know what a cat looks like; hence, the model might generate
> a dog napping. We can reduce this epistemic uncertainty (from lack
> of knowledge) by collecting more training data.
>
>
> [1] Eyke Hüllermeier and Willem Waegeman. Aleatoric and epistemic
> uncertainty in machine learning: An introduction to concepts and
> methods. Machine learning, 110(3):457–506, 2021.

---

> > ### Author Response · Authors · 2025-11-23
> > **Response to Reviewer Mt95 (Part II)**
> >
> > **Clarification on the implementation of our method.** Based on
> > the preceding definitions of aleatoric and epistemic uncertainty, we
> > note that aleatoric uncertainty does **not** depend on the model, while
> > epistemic uncertainty does.
> > We acknowledge that practical limitations often hinder a perfect
> > implementation of theoretically-grounded frameworks. However,
> > we clarify that our implementations provide useful estimates of
> > aleatoric uncertainty and epistemic uncertainty. Specifically, aleatoric
> > uncertainty is encapsulated by the variability in the task. Prior work
> > has extensively demonstrated the impressive generation capabilities
> > of large language models (LLMs). More importantly, LLMs were
> > trained on internet-scale human data, making them effective language
> > generators that closely align with human intent. We use LLMs
> > as an effective means of capturing the variability inherent in the
> > prompt (i.e., in the task specification), by sampling multiple possible
> > instances of the task from the LLM. This variability essentially
> > defines the aleatoric uncertainty associated with the task. Considering
> > the previous example, two possible variations of the task include:
> > (1) “a cat napping on a purple bed in the back of a pickup truck on
> > a **cloudy** day”, and (2) “a cat napping on a purple bed in the back
> > of a pickup truck on a **sunny** day.” Although LLMs do not exactly
> > match the true distribution of compatible tasks, this gap is small in
> > practice.
> > Likewise, we emphasize that latent variable modeling is a well-
> > studied probabilistic modeling approach that has been shown to
> > be effective in vision and natural language applications. Extensive
> > research has demonstrated that high-dimensional features can often
> > be decomposed into a small set of lower-dimensional factors, without
> > a significant loss of expressiveness or information. Hence, we use
> > this technique as a practical means of analyzing the distribution
> > of generated videos. We clarify that this approach leads to some
> > loss of information, like all compression-based methods; however,
> > the resulting information loss is generally minimal. In essence, our
> > method provides an effective practical framework for quantifying
> > the uncertainty of video models while minimizing the gap between
> > theory and implementation.
> >
> > **Clarification on Assumptions in our Method.** We highlight
> > important nuances in our method. First, we clarify that there is no
> > independence assumption between the text prompt and the generated
> > video. In fact, we directly condition the generated video on the (latent)
> > text prompt. It appears that there might be some confusion between
> > *independence* which our paper does not assume and *conditional
> > independence* between the video and text prompt which our paper
> > uses. We note that this conditional independence assumption is
> > valid because the actual generated video is always conditioned on a
> > specific text input and is conditionally independent of all other text
> > inputs.
> > Second, we clarify that entropy is associated with a probability
> > space, which is over a latent variable in our paper. However, we
> > emphasize that there is no geometric distortion since we do not have
> > a reference probability space for computing entropy that is different
> > from the latent variable probability space. More concretely, the
> > embedding dimensions and projection define the random variables
> > in the probability space, and the entropy is computed with respect
> > to this space with no distortion. Lastly, we demonstrate through
> > extensive experiments that the uncertainty estimates computed by our
> > methods are interpretable both quantitatively and qualitatively, e.g.,
> > in Figures 3, 4, and 5. Moreover, in Appendix C, we have included
> > more experiments in our paper to further show the effectiveness of
> > our method.

---

> > > ### Author Response · Authors · 2025-11-23
> > > **Response to Reviewer Mt95 (Part III)**
> > >
> > > **Clarification on Experimental Protocols.** We clarify that prior
> > > work [2, 3] has extensively used CLIP-based metrics in evaluating
> > > the instruction following and frame consistency performance of video
> > > models. Hence, our choice of this metric is not a random choice.
> > > Nevertheless, through our experiments, we verify that the CLIP score
> > > metric actually aligns with human judgment. Concretely, we find that
> > > the CLIP score is strongly negatively correlated with human-ranked
> > > uncertainty at statistically significant levels. This result is consistent
> > > with prior work and further aligns with the intuition that humans
> > > assess video consistency in vague video generation tasks primarily
> > > from a visual-semantic perspective.
> > > Likewise, we clarify that we do not randomly set one component
> > > of the total uncertainty to zero to evaluate the other component.
> > > Rather, based on the decomposition of total uncertainty into aleatoric
> > > uncertainty and epistemic uncertainty, we note that total uncertainty
> > > equals aleatoric uncertainty, when the epistemic uncertainty is close
> > > to zero. The converse also holds. Given this fact, we examine the
> > > correlation of the aleatoric uncertainty with accuracy by selecting
> > > the samples with the lowest epistemic uncertainty, and repeat the
> > > procedure for epistemic uncertainty. Although this analysis does
> > > not perfectly recover each component, it provides valuable insight
> > > into the calibration of these components, as demonstrated in our
> > > experiments. We have highlighted these points in our revised paper.
> > >
> > > We hope our response has clarified the theoretical grounding of
> > > our paper and the tradeoffs between perfect theoretical realizations
> > > and practical implementations. We would be glad if you could update
> > > your rating to reflect these changes.
> > >
> > >
> > > [1] Eyke Hüllermeier and Willem Waegeman. Aleatoric and epistemic
> > > uncertainty in machine learning: An introduction to concepts and
> > > methods. Machine learning, 110(3):457–506, 2021.
> > >
> > > [2] Team Wan, Ang Wang, Baole Ai, Bin Wen, Chaojie Mao, Chen-Wei
> > > Xie, Di Chen, Feiwu Yu, Haiming Zhao, Jianxiao Yang, et al. Wan:
> > > Open and advanced large-scale video generative models. arXiv
> > > preprint arXiv:2503.20314, 2025.
> > >
> > > [3] Jack Hessel, Ari Holtzman, Maxwell Forbes, Ronan Le Bras, and
> > > Yejin Choi. Clipscore: A reference-free evaluation metric for image
> > > captioning. In Proceedings of the 2021 conference on empirical
> > > methods in natural language processing, pages 7514–7528, 2021.

---

> ### Comment · Reviewer_Mt95 · 2025-11-26
>
> Thank you to the authors for their detailed rebuttal.
>
> Recent work shows that it is often not possible to cleanly decompose total uncertainty into “aleatoric” and “epistemic” components in modern neural networks [1] (especially those that take in text-based input). Instead the two components produced by standard decompositions can be highly correlated and strongly model-dependent rather than reflecting independent, intrinsic sources of uncertainty.  To illustrate this, Kirchhof et al. [2] survey multiple schools of thought and emphasize that what is labeled “aleatoric” versus “epistemic” conflicts across formulations, and in practice depends on the chosen model class, inductive biases, and somewhat arbitrary choices around what is considered data noise and what is considered model misspecification.  This is not to say that these notions are not useful, just that I believe a more nuanced and rigorous discussion is necessary.  The place where this really sticks out to me is teh definition of aleatoric uncertainty in line 178.
>
> While I believe that there is a significant opportunity to open the discussion around UQ in video generation models, I find that the work (as well as the author response to my review) does not put forth a careful or rigorous treatment of the subject.  Your example that “a cat napping on a purple bed in the back of a pickup truck” has aleatoric uncertainty w.r.t. time of day, weather, or cat appearance illustrates this issue: these features are not inherently random; they are only random with respect to some underlying distribution of interest. These would not be significant sources of aleatoric uncertainty if, for example, the distribution of interest only contains daytime, rainy scenes with a specific breed of cat.
>
> I also am concerned that the sole reference [3] used to rebut my main cited weakness is not cited in the original or revised manuscript.
>
> I will maintain my score.
>
>
> - [1] Benchmarking Uncertainty Disentanglement: Specialized Uncertainties for Specialized Tasks (https://arxiv.org/abs/2402.19460)
> - [2] Position: Uncertainty Quantification Needs Reassessment for Large-language Model Agents https://arxiv.org/abs/2505.22655
> - [3] Eyke Hüllermeier and Willem Waegeman. Aleatoric and epistemic uncertainty in machine learning: An introduction to concepts and methods. Machine learning, 110(3):457–506, 2021.

---

### Official Review · Reviewer_nQDP · 2025-10-28

**Soundness:** 3
**Presentation:** 4
**Contribution:** 4
**Rating:** 8
**Confidence:** 4

**Summary:**

- The authors introduced a framework to measure the uncertainty of video generative models.
- The framework consists of a metric for evaluating the calibration of video models based on robust rank correlation estimation.
- They also introduce S-QUBED, a black-box UQ method for video models. S-QUBED effectively distinguishes between uncertainty arising from ambiguous prompts and uncertainty stemming from the model's lack of knowledge.
- They will also release a dataset of 40K videos across diverse tasks to help benchmark calibration in video models.
- The authors used their method to disentangle and understand aleatoric and epistemic misunderstandings of the video generation models. For example, to assess epistemic misunderstanding, they generated multiple videos for the same prompt and embedded them. Then, they measured the embeddings' spread, with wider spread indicating higher epistemic uncertainty.
- For the main result of their work, they further study the correlation between accuracy and the different uncertainties. They find that when uncertainty is higher, accuracy tends to be lower. This holds for both overall uncertainty and aleatoric/epistemic misunderstanding.

**Strengths:**

- Uncertainty quantification of LLMs is well studied, but not studied at all for video generation models. This work was novel in that it studied uncertainty quantification of video generation models.
- The black box approach makes it accessible to evaluate any video generation model.
- The authors presented the material well, providing the necessary background to understand the motivation and importance of this work, which is especially important given its novelty.

**Weaknesses:**

- I would like to see empirical results and to validate S-QUBED on other open (non-API) video models, given that it is a black-box approach. The authors mentioned that different models were considered but not evaluated due to access and compute constraints. However, I believe there should be multiple open text-to-video models to evaluate S-QUBED on (e.g., OpenSora).
- Typical metrics (e.g., CLIP, PSNR) for evaluating text-to-image and text-to-video models often do not align with human judgment. Would like to see the correlation of uncertainty with human judgment metrics.

**Questions:**

No questions as the background, motivation, and results were presented well.

---

> ### Author Response · Authors · 2025-11-23
> **Response to Reviewer nQDP**
>
> Thank you for recognizing the novelty and applicability of our work.
> We are grateful for your valuable feedback, which we found to
> be really helpful and constructive. We group our response into the
> following sections:
>
> **Additional Experiments on Other Video Models.** In Appendix C,
> we have included additional evaluations on three state-of-the-art
> video models: OpenSora, CogVideoX, and Veo 3, covering both
> open-source and closed-source models with varying video generation
> performance. In contrast to Cosmos-Predict2 which uses a text-
> to-image-to-video pipeline, these models use a more unified text-
> to-video pipeline. Further, using these models, we validate the
> practicality of our method when implementation challenges arise.
> For example, Veo 3 imposes strict cap on the number of videos
> that a user can generate, in addition to the significant cost of each
> generation task. Likewise, OpenSora requires large-memory GPUs
> such as the A100 and H100 GPUs, which could also pose a practical
> limitation. We evaluate the calibration of our method with these
> models across different hyperparameter choices (e.g., number of
> latent prompts and number of generated videos per latent prompt)
> to validate the practicality of our method under the constraints
> imposed by these video models. Specifically, we run experiments
> with fewer latent prompts and generated videos per latent prompt to
> minimize costs and stay below the API limit, e.g., with Veo 3. We
> find that S-QUBED computes well-calibrated uncertainty estimates
> for all models, even with very few generated videos. We observe a
> strong negative correlation between the estimated uncertainty and
> accuracy, further demonstrated by visualizations of the ground-truth
> and generated videos. In summary, we show that S-QUBED provides
> a practical, well-calibrated method for uncertainty quantification
> of text-to-image-to-video and text-to-video models with different
> compute or cost constraints.
>
> **Clarification on Metric and Human Judgment Alignment.** We
> agree with your comment. Hence, we evaluated the alignment of
> these standard metrics (CLIP, PSNR, SSIM, and LPIPS) with human
> judgment. Specifically, we computed the correlation between each
> of these metrics and human-ranked uncertainty, shown in Figure 3.
> We find that the the CLIP score metric is well-aligned with human
> assessment of uncertainty at statistically significant levels, unlike
> the perceptual metrics, such as PSNR, SSIM, and LPIPS. This
> finding aligns with the intuition that humans assess video consistency
> in vague video generation tasks primarily from a visual-semantic
> perspective, which the perceptual metrics fail to capture. We have
> highlighted these points in the Appendix.
>
>
> We appreciate your deeply insightful comments. We have revised
> our paper to address your concerns and would be glad if you could
> update your rating to reflect the changes to the paper.

---

### Official Review · Reviewer_RL8t · 2025-10-30

**Soundness:** 3
**Presentation:** 3
**Contribution:** 2
**Rating:** 6
**Confidence:** 2

**Summary:**

This paper is (to the authors’ knowledge) the **first study of uncertainty quantification (UQ) for text-to-video models**, proposing a three-part framework: (i) a **calibration metric** based on robust rank correlation between uncertainty and task accuracy, (ii) a black-box UQ method, **S-QUBED**, that uses a **latent-space factorization** to **decompose total predictive uncertainty** into **aleatoric** (prompt vagueness) and **epistemic** (model ignorance) components, and (iii) a ~**40K-video UQ dataset** for benchmarking. Experiments on VidGen-1M and Panda-70M show that S-QUBED’s total uncertainty is **significantly negatively correlated** with semantic accuracy (CLIP score), and its decomposition yields calibrated aleatoric/epistemic trends on subsets where the other source of uncertainty is minimal.

**Strengths:**

* Positions UQ for video generation as a first-class problem; formal **entropy decomposition** (h(V|\ell)=h(V|Z)+h(Z|\ell)) cleanly maps to epistemic vs. aleatoric sources.
* **S-QUBED** operates without model internals, aligning with many **closed-source video models**.
* Uses **Kendall’s τ** and demonstrates **significant negative correlation** between S-QUBED uncertainty and **CLIP accuracy**, with visuals that match the trend.
* Empirical **disentangling** of aleatoric vs. epistemic uncertainty shows expected behavior on curated subsets.
* Plans to release a **~40K-video UQ dataset** covering diverse tasks.

**Weaknesses:**

* Calibration hinges primarily on **CLIP similarity**; other perceptual metrics (SSIM/PSNR/LPIPS) show weak or insignificant correlations, raising concerns about **metric sensitivity** and potential semantic-evaluator bias.
* Estimating **epistemic uncertainty** requires **multiple generations per latent prompt**, which the authors acknowledge as a limitation.
* Main experiments use **Cosmos-Predict2** and two datasets; broader **model diversity** and real-world perturbations (codecs, length, audio conditions) are not deeply explored.

**Questions:**

1. Beyond CLIP, what **additional accuracy signals** (e.g., human semantic judgments, video-text retrieval scores, physics consistency probes) are necessary to **validate calibration** and mitigate evaluator bias?
2. What **sampling schedules** (fewer latent prompts/videos, adaptive stopping) or **latent-space proxies** would you require to deem S-QUBED **computationally practical** without sacrificing epistemic resolution?
3. Which **additional models/datasets** or **deployment artifacts** (compression, prompt styles, audio/no-audio) would most convincingly demonstrate that the **aleatoric/epistemic decomposition** remains **stable and calibrated** in the wild?

---

> ### Author Response · Authors · 2025-11-23
> **Response to Reviewer RL8t (Part I)**
>
> Thank you for recognizing the rigor and applicability of our work.
> We are grateful for your valuable feedback, which we found to
> be really helpful and constructive. We group our response into the
> following sections:
>
> **Why CLIP metric?** We acknowledge that suitable metrics for
> evaluating video quality and consistency still remain an open research
> question [1, 2]. We believe that the suitability of a metric strongly
> depends on the application context. In our case, we are interested in
> the alignment between the generated and ground-truth videos in the
> dataset, particularly in the semantic space. We note that the vagueness
> of many video generation tasks in our work inherently results in a
> notable divergence between the ground-truth and generated videos in
> the visual space. As a result, quantifying alignment with perceptual
> metrics like SSIM, PSNR, and LPIPS does not provide a reliable
> measure of consistency between the generated and ground-truth
> videos. This fact explains the lack of significant correlation between
> these metrics and human-ranked uncertainty in the results in our
> paper (Figure 3). In contrast, semantic metrics capture precisely this
> information, abstracting away smaller details that would otherwise
> derail meaningful comparisons while preserving the core details
> of the task prompt that is visualized in the ground-truth video.
> Based on this fundamental point, recent works have used CLIP
> similarity to assess instruction following and frame consistency
> metrics [2, 3]. Although video-text retrieval scores, VLM-based
> scores, and physics consistency probes can all provide valuable
> accuracy signals, these metrics often introduce additional challenges,
> e.g., hallucinations, limiting their effectiveness. We believe that
> alignment with human semantic judgment is useful. Hence, in our
> experiments, we verify that the CLIP score metric actually aligns
> with human judgment. Concretely, we find that the CLIP score
> is strongly negatively correlated with human-ranked uncertainty at
> statistically significant levels. This finding aligns with the intuition
> that humans assess video consistency in vague video generation tasks
> primarily from a visual-semantic perspective. We have highlighted
> these points in our revised submission.
>
> **Epistemic Uncertainty Estimation Method.** We acknowledge that
> generating multiple videos might pose a limitation. However, we
> emphasize that this limitation is not unique to our method. Moreover,
> in practical situations, this limitation turns out to be non-prohibitive.
> In fact, our method is highly flexible, making it adaptable to broad
> range of task settings, e.g., situations with cost or API constraints.
> In Appendix C, we demonstrate the effectiveness of our method
> with fewer number of latent prompts and fewer generated videos
> per latent prompt, showing that S-QUBED remains well-calibrated
> in these settings. These results underscore the practicality of our
> method and its applicability to video models with stringent cost or
> generation constraints, such as Veo 3 and OpenSora. We summarize
> the different hyperparameters for the number of latent prompts and
> number of generated videos per latent prompt in Table I.
>
> | Video Model | Number of Latent Prompts | Number of Videos per Latent Prompt |
> |-----------------|----|----|
> | Cosmos-Predict2 | 10 | 10 |
> | CogVideoX       | 10 | 10 |
> | Veo 3           | 5  | 4  |
> | OpenSora        | 3  | 3  |
>
>
> [1] Yaofang Liu, Xiaodong Cun, Xuebo Liu, Xintao Wang, Yong
> Zhang, Haoxin Chen, Yang Liu, Tieyong Zeng, Raymond Chan, and
> Ying Shan. Evalcrafter: Benchmarking and evaluating large video
> generation models. In Proceedings of the IEEE/CVF Conference
> on Computer Vision and Pattern Recognition, pages 22139–22149,
> 2024.
>
> [2] Team Wan, Ang Wang, Baole Ai, Bin Wen, Chaojie Mao, Chen-Wei
> Xie, Di Chen, Feiwu Yu, Haiming Zhao, Jianxiao Yang, et al. Wan:
> Open and advanced large-scale video generative models. arXiv
> preprint arXiv:2503.20314, 2025.
>
> [4] Jack Hessel, Ari Holtzman, Maxwell Forbes, Ronan Le Bras, and
> Yejin Choi. Clipscore: A reference-free evaluation metric for image
> captioning. In Proceedings of the 2021 conference on empirical
> methods in natural language processing, pages 7514–7528, 2021.

---

> > ### Author Response · Authors · 2025-11-23
> > **Response to Reviewer RL8t (Part II)**
> >
> > **Further Experiments with More Video Models and Deployment
> > Artifacts.** In Appendix C, We have included additional evaluations
> > on three state-of-the-art video models: OpenSora, CogVideoX, and
> > Veo 3, covering both open-source and closed-source models. In
> > contrast to Cosmos-Predict2 which uses a text-to-image-to-video
> > pipeline, these models use a more unified text-to-video pipeline.
> > Likewise, these models generate videos of different lengths and
> > video resolution. Moreover, while some models like Veo 3 generate
> > audio, others such as OpenSora and CogVideoX do not generate
> > audio. Further, using these models, we validate the practicality of
> > our method when implementation challenges arise. For example,
> > Veo 3 imposes a strict cap on the number of videos that a user can
> > generate, in addition to the significant cost of each generation task.
> > Likewise, OpenSora requires large-memory GPUs such as the A100
> > and H100 GPUs, which could also pose a practical limitation. We
> > evaluate the calibration of our method with these models across
> > different hyperparameter choices (e.g., number of latent prompts
> > and number of generated videos per latent prompt) to validate
> > the practicality of our method under the constraints imposed by
> > these video models. Specifically, we run experiments with fewer
> > latent prompts and generated videos per latent prompt to minimize
> > costs and stay below the API limit, e.g., with Veo 3. We find
> > that S-QUBED computes well-calibrated uncertainty estimates for
> > all models, even with very few generated videos. We observe a
> > strong negative correlation between the estimated uncertainty and
> > accuracy, further demonstrated by visualizations of the ground-truth
> > and generated videos. In summary, we show that S-QUBED provides
> > a practical, well-calibrated method for uncertainty quantification of
> > text-to-image-to-video and text-to-video models across different test-
> > time conditions/constraints.
> >
> > We appreciate your valuable comments on our work. We have
> > revised our paper to address your concerns and would be glad if
> > you could update your rating to reflect the changes to the paper.
> >
> >
> > [1] Yaofang Liu, Xiaodong Cun, Xuebo Liu, Xintao Wang, Yong
> > Zhang, Haoxin Chen, Yang Liu, Tieyong Zeng, Raymond Chan, and
> > Ying Shan. Evalcrafter: Benchmarking and evaluating large video
> > generation models. In Proceedings of the IEEE/CVF Conference
> > on Computer Vision and Pattern Recognition, pages 22139–22149,
> > 2024.
> >
> > [2] Team Wan, Ang Wang, Baole Ai, Bin Wen, Chaojie Mao, Chen-Wei
> > Xie, Di Chen, Feiwu Yu, Haiming Zhao, Jianxiao Yang, et al. Wan:
> > Open and advanced large-scale video generative models. arXiv
> > preprint arXiv:2503.20314, 2025.
> >
> > [4] Jack Hessel, Ari Holtzman, Maxwell Forbes, Ronan Le Bras, and
> > Yejin Choi. Clipscore: A reference-free evaluation metric for image
> > captioning. In Proceedings of the 2021 conference on empirical
> > methods in natural language processing, pages 7514–7528, 2021.

---

### Official Review · Reviewer_eJzE · 2025-11-01

**Soundness:** 3
**Presentation:** 3
**Contribution:** 3
**Rating:** 6
**Confidence:** 4

**Summary:**

The paper proposes a black-box framework that lets text-to-video models express uncertainty by decomposing predictive uncertainty into aleatoric (prompt vagueness) and epistemic (model ignorance) components. The framework is evaluated with a rank-correlation-based calibration metric, and a 40K-video UQ benchmark is released.

**Strengths:**

1. The paper is well-presented, well-written, and the motivation is justified.
2. The research topic of principled evaluation of synthetic videos is very timely and important.
3. The proposed dataset will be valuable.

**Weaknesses:**

1. The method’s evidence of **general** video-model UQ almost entirely depends on one text-to-image-to-video pipeline (Cosmos-Predict2). While I appreciate that authors state the API/compute constraints, it will be more convincing if the paper proposes potential solutions or fixes to overcome the challenge. That being said, the practicality and calibration of stronger video models shall be evaluated.
2. Please fix salient typos such as "video modes" (Page 3) and "peak signal-to-noise ration" (Page 13).

**Questions:**

While there are several weaknesses stated above, I believe this paper will be contributive and will provide new insights to the community. I therefore have the initial rating of 6 for this paper. Please note that my final rating will be conditioned on the soundness of the rebuttal.

---

> ### Author Response · Authors · 2025-11-23
> **Response to Reviewer eJzE**
>
> Thank you for recognizing the timeliness and value of our work. We
> are grateful for your valuable feedback, which we found to be really
> helpful and constructive. We group our response into the following
> sections:
>
> **Additional Experiments on Other Video Models.** In Appendix C,
> we have included additional evaluations on three state-of-the-art
> video models: OpenSora, CogVideoX, and Veo 3, covering both
> open-source and closed-source models with varying video generation
> performance. In contrast to Cosmos-Predict2 which uses a text-
> to-image-to-video pipeline, these models use a more unified text-
> to-video pipeline. Further, using these models, we validate the
> practicality of our method when implementation challenges arise.
> For example, Veo 3 imposes a strict cap on the number of videos
> that a user can generate, in addition to the significant cost of each
> generation task. Likewise, OpenSora requires large-memory GPUs
> such as the A100 and H100 GPUs, which could also pose a practical
> limitation. We evaluate the calibration of our method with these
> models across different hyperparameter choices (e.g., number of
> latent prompts and number of generated videos per latent prompt) to
> validate the practicality of our method under the constraints imposed
> by these video models. Specifically, we run experiments with fewer
> latent prompts and fewer generated videos per latent prompt to
> minimize costs and stay below the API limit, e.g., with Veo 3. We
> find that S-QUBED computes well-calibrated uncertainty estimates
> for all models, even with very few generated videos. We observe a
> strong negative correlation between the estimated uncertainty and
> accuracy, further demonstrated by visualizations of the ground-truth
> and generated videos. In summary, we show that S-QUBED provides
> a practical, well-calibrated method for uncertainty quantification
> of text-to-image-to-video and text-to-video models with different
> compute or cost constraints.
>
> **Typos.** We have fixed the typos.
>
> We appreciate your understanding that our work provides value to
> the community. We have revised our paper to address your concerns
> and would be glad if you could update your rating to reflect the
> changes to the paper

---

> > ### Comment · Reviewer_eJzE · 2025-11-27
> >
> > Thank authors for the additional information provided. I went to check the mentioned section and found that it indeed addresses my concerns. Yet, I also partially agree with the review from Mt95 that the technical novelty of the paper is somehow limited. I personally enjoy the presentation of the paper (a great motivation with well-crafted visualization and math framework to clearly convey the idea). This makes this paper an outstanding one in my review batch. I will maintain the final rating of 6.

---

### Meta-Review · Area_Chair_Msvp · 2026-01-03

**Summary:**

The paper studies the question of uncertainty estimation in video-generation models. The authors propose a separation of the uncertainty into two components: epistemic and aleatoric.

The paper considers a pipeline where the given prompt is first converted to a detailed textual prompt, and then a video is generated based on a detailed prompt. The authors equate aleatoric uncertainty with the uncertainty in the detailed textual prompt, while the epistemic uncertainty is the uncertainty in the video conditioned on the detailed prompt.

The authors then propose a method for estimating both of these uncertainty types by the same strategy: embed using an embedding model to a collection of samples, fit a VMF distribution, and compute its entropy.

The authors then show that this uncertainty metric is correlated with accuracy of instruction following for a collection of video generation models and prompts.

Reviewer Mt95 identifies an important issue: the proposed split into epistemic and aleatoric uncertainty is not well-justified. Indeed, it is unclear why the aleatoric uncertainty is captured by the uncertainty in the detailed prompt. Aleatoric uncertainty should capture the variability of the samples conditional on the original prompt in a perfectly trained video model on a given data distribution. It seems likely that at least some of this variability will not be verbalized by an LLM when asked to expand a given prompt: the patterns of the movement of the leaves of a tree in the wind, the shapes of the clouds, etc. Even in the example of the figure 1, it is suggested that the person in the video corresponds to the epistemic uncertainty, while the lighting of the beach corresponds to aleatoric uncertainty; it is unclear why that should be the case, and why the appearance of "Jeff Einstein" is something that should count as epistemic uncertainty.

Moreover, the proposed notion of aleatoric uncertainty does not depend on the data distribution that we would train our model on, which is conceptually problematic. If all of the videos we train on are just "black screen for 10 seconds" regardless of the prompt, the method would still estimate high aleatoric uncertainty for vague prompts, even though the actual videos would all look the same regardless of the prompt, so there is no aleatoric uncertainty.

So to sum up, the proposed distinction between the uncertainty types is not well-justified, and does not correspond to the standard notions of epistemic and aleatoric uncertainty in general. The estimates computed by the method may still be useful, e.g. in identifying vague prompts. However, I believe that positioning the paper as a way of measuring epistemic and aleatoric uncertainty in video generation is incorrect.

The paper has multiple strengths, e.g. the authors evaluate multiple models across diverse tasks; they also plan to release a dataset targeting UQ in video generation. Generally, I believe this paper could be useful to the practitioners, but I believe the framing needs to be updated.

**Reviewer Concerns:**

The concern raised by the reviewer Mt95 which I described above was not addressed by the rebuttal. The reviewer responded to the rebuttal and stated that the concern was not addressed.

Other reviewers raised concerns about the choice of models, which was addressed in the rebuttal: the authors added other models.

Another concern was with the choice of the ClipScore as a metric. The authors provided a justification for their choice, which partially addresses the concern.

**Reviewer Scores:**

Reviewer eJzE: 6 -> 6 (responded to rebuttal)
Reviewer RL8t: 6 -> 6 (guess)
Reviewer nQDP: 8 -> 8 (guess)
Reviewer Mt95: 2->2 (responded to rebuttal)

---

### Decision · Program_Chairs · 2026-01-26

Reject